# Endogenous Retroviruses and Placental Evolution, Development, and Diversity

**DOI:** 10.3390/cells11152458

**Published:** 2022-08-08

**Authors:** Kazuhiko Imakawa, Kazuya Kusama, Tomoko Kaneko-Ishino, So Nakagawa, Koichi Kitao, Takayuki Miyazawa, Fumitoshi Ishino

**Affiliations:** 1Research Institute of Agriculture, Tokai University, Kumamoto 862-8652, Japan; 2Department of Endocrine Pharmacology, Tokyo University of Pharmacy and Life Sciences, Tokyo 192-0392, Japan; 3School of Medicine, Tokai University, Tokyo 259-1193, Japan; 4Department of Molecular Life Science, Tokai University School of Medicine, Nakagawa 259-1193, Japan; 5Laboratory of Virus-Host Coevolution, Institute for Life and Medical Sciences, Kyoto University, Kyoto 606-8507, Japan; 6Institute of Research, Tokyo Medical and Dental University, Tokyo 113-8510, Japan

**Keywords:** placenta, structural diversity, endogenous retrovirus (*ERV*), mammals

## Abstract

The main roles of placentas include physical protection, nutrient and oxygen import, export of gasses and fetal waste products, and endocrinological regulation. In addition to physical protection of the fetus, the placentas must provide immune protection throughout gestation. These basic functions are well-conserved; however, placentas are undoubtedly recent evolving organs with structural and cellular diversities. These differences have been explained for the last two decades through co-opting genes and gene control elements derived from transposable elements, including endogenous retroviruses (ERVs). However, the differences in placental structures have not been explained or characterized. This manuscript addresses the sorting of ERVs and their integration into the mammalian genomes and provides new ways to explain why placental structures have diverged.

## 1. Introduction

It is well accepted that the transition from oviparity to viviparity occurred independently in many animal lineages. Along with two other lizard species, *Lerista bougainvillii* and *Zootoca vivipara*, the yellow-bellied three-toed skink, *Saiphos equalis*, exhibits intraspecific variation in which both oviparity and viviparity exist [1,2]. Although many genes are expressed throughout their reproductive cycles, the differentially expressed genes within oviparous and viviparous individuals have surprisingly similar biological functions important for sustaining embryos [3]. Although viviparity is found not only in lizards but also in fishes, amphibians and other reptiles [4,5], the cell layer known as the trophectoderm has not been found in these animal species. In most animals, life begins with fertilization of the egg with sperm, followed by mitotic cell divisions or cleavages to generate blastomeres, from which the primary germ layers, namely endoderm, mesoderm and ectoderm, are generated. These cell layers eventually give rise to the formation of the various tissues essential for animal life.

The initial development of placental mammals differs from other members of the animal kingdom, in which a blastocyst, consisting of the trophectoderm (TE) and the inner cell mass (ICM), is initially formed; TE becomes the major cell type for the placenta, and ICM forms the body structure, including the amnion, which directly surrounds the fetal body. The blastocyst must implant in the maternal endometrium, after which the three germ layers form from the ICM. TE cell layers are epithelial in nature, each expressing Ca^2+^-dependent adhesion molecule E-cadherin [6,7]. A typical epithelium also possesses zonula occluden (ZO) proteins, which connect cell membranes between the apposing cells through tight junctions (TJ) [8,9]. 

For the past two decades, various new technologies to analyze DNA and RNA sequences have become available. In addition, various bioinformatics programs have also been developed to analyze the repetitive and non-repetitive sequences on the genomes, which enable the detection of DNA sequences originating from integrated retroviral sequences. However, mammalian placentas’ evolution and structural diversity have not been fully unraveled. This review endeavors to explain the origin and diversity of mammalian placentas.

## 2. Structural Diversity of Mammalian Placentas

Following fertilization, the first differentiation of embryonic cells results in the formation of a blastocyst with an inner cell mass (ICM) and an outer trophectoderm. The ICM differentiates into the embryo, amnion, yolk sac, and allantois, while the trophectoderm develops into the chorion, the fetal contribution to the placenta. In most mammals, the yolk sac is displaced with the allantois, forming chorioallantoic placentation [10]. In rodents and lagomorphs, however, the yolk sac forms the so-called inverted yolk sac placenta with a maternal-facing absorptive epithelium, which persists until term [11]. Inverted yolk sac placenta or yolk sac predominance could be the form of placentation in early mammals, which are later replaced with chorioallantoic placentation. The latter is classified into diffuse (pigs and horses), cotyledonary (ruminants), zonary (dogs and cats) and discoids (murines and primates) (Figure 1). Placental cell layer separations are those of epitheliochorial, endotheliochorial, and hemochorial (Figure 2).

An epitheliochorial placenta is found in several orders of animals, including even-toed ungulates, whales, dolphins and lower primates, of which the outer chorioallantoic membrane is in direct contact with the maternal uterine epithelium. The synepitheliochorial placenta is found exclusively in ruminants. As the uterine epithelium is lost, the trophectoderm is directly apposed to the maternal connective tissue, and the placentomal (cotyledonary) chorioallantoic placenta develops at discrete regions of pre-existing, non-glandular caruncular areas on the uterine epithelium [12]. An endotheliochorial placenta is seen in carnivores such as cats and dogs; however, this type of placenta is also found in distantly related elephants [13]. In this type of placentation, the maternal capillaries are located close to the trophectoderm cells of the chorioallantoic membrane, resulting from stromal thinning and a loss of uterine epithelium.

It should be noted that hyenas belong to the order carnivore, but the hyena undergoes hemochorial placentation (Figure 3). A hemochorial placentation is seen in higher primates, including humans and many rodents—the latter in which the inverted yolk sac co-exists. In this type of placentation, maternal blood is directly in touch with the trophectoderm cells and is thus unlike other types of placentation. The capillary endothelium does not exist at or near the region where metabolic exchange occurs.

## 3. Cells Gain Their Mobility through Epithelial-Mesenchymal Transition (EMT)

The outer layer of trophectodermal cells is basically epithelial cells because they possess apicobasal cell polarity, lateral junctions with neighboring cells, and basal contact with the basement membranes [40,41]. These cells, however, still manage to adhere to the uterine epithelium through its apical domain, beginning periods of peri-implantation. Unless trophectodermal cells undergo an epithelial–mesenchymal transition (EMT) [42,43,44,45], these cells do not have the ability of both cell migration and cell adhesion to the maternal cells. Thus, EMT is generally coupled to the acquisition of migratory and invasive properties in the invasive mode of trophectodermal cells.

When DNA synthesis takes place in the absence of karyo- and cytokinesis, giant trophoblast cells arise through endoreduplication, as in the murine species. Except for animal species exhibiting epitheliochorial placentation, such as pigs and horses, the outermost layer of trophectodermal cells in many eutherians go through cell fusion, resulting in the formation of bi-nucleated, multi-nucleated or syncytiotrophoblast cells [46]. Like malignant cells, trophectodermal cells can invade neighboring cells; however, the multi-nucleated or syncytiotrophoblast cells do not go through cell cycles; thus, their invasiveness to neighboring cells is limited [47].

Unlike in murine and primate species, the bovine blastocyst does not invade the uterine endometrium. Instead, it elongates beginning on days 13–14 (day 0 = day of estrus), and as the elongation subsides on days 19–20, the elongated bovine conceptus begins its attachment to the uterine epithelium. Two to three days later, the bovine conceptus starts its adhesion to the uterine epithelium when conceptus trophectoderm expresses mesenchymal markers, such as N-cadherin (CDH2) and vimentin (VIM) [48]. It should be noted that bi-nucleated cells start to appear on day 20, and the number of multi-nucleated cells increases on day 22 when bovine trophectodermal cells express transcription factors *SNAI2*, *ZEB1*, *ZEB2*, *TWIST1*, *TWIST2*, and *KLF8* transcripts [48]. On day 22, bovine trophectodermal cells with EMT marker expression undergo both cell fusion and cell adhesion, suggesting that EMT-related factor expression is required for the proper adhesion of elongated conceptus trophectodermal cells to the maternal endometrium.

The acquisition of EMT in the non-invasive mode of trophectodermal cells was demonstrated in the following experiments. Using bovine trophoblast CT-1 cells [49], bovine endometrial epithelial cells (bEECs) [50] and those of ex vivo samples. For EMT-related molecule expression, a transcription factor OVOL2 is required to be down-regulated on day 22 [51], whereas the EMT markers induced by day 22, including uterine flushing (UF) or activin A, are inhibited by follistatin (FST) in the CT-1 and bEEC cells co-culture system [52]. The treatment with activin A or day 22 UF in co-cultured CT-1 cells induced cell characteristics similar to those of EMT [52], involving the down-regulation of apicobasal polarity, loss of cell–cell adhesion and expression of MMPs [53]. It should be noted that unique to ruminant ungulates, bi-nucleated cells fuse heterotypically with uterine epithelial cells, resulting in the trinucleated cell formation found in the uterine endometrium [54]. Seo and coworkers provided evidence that ovine trophectodermal cells went through homotypic cell fusion, not heterotypic, and formed syncytiotrophoblast-like fused cells [55]. This observation agrees with our recent publication in which bi-nucleated and multi-nucleated cells observed in vivo are all homotypic cell fusions [56].

## 4. Formation of Syncytiotrophoblast Layer in the Mammalian Species

The presence and importance of syncytiotrophoblast cells are often overlooked, particularly the differences in their cellular structures. These cells are derived from trophectodermal cell fusion and are located next to the maternal cell components. These cells manage efficient nutrient and gas exchange. Because syncytiotrophoblast cells are located next to the maternal cells, they are involved in the immunotolerance of the conceptus by the maternal immune system [57,58].

The chorionic membrane consists of mono-, di- or tri-layers of trophoblast cells. In hemochorial placentation, the human placenta, discoidal in shape, is classified as a monochorial interface as well as exhibiting villous maternal–fetal interdigitations. In murines, including Muridae and Cricetidae, a labyrinthine placenta is developed in the hemotrichorial organization with three trophoblast layers [59]. The first layer closest to the maternal stroma and/or decidua consists of mononucleate cytotrophoblast cells, and the second layer consists of a syncytiotrophoblast cell layer, whereas the third trophoblast layer consists of cyotrophoblast cells [60]. 

On the other hand, beavers, rabbits, and bats have two layers of chorionic membranes, or dichorial placentas, with the first layer consisting of syncytiotrophoblast cells and the second of cytotrophoblast cells. It should be noted that Provaviidae (Hyrax), Tenrec ecaudatus and Dipodidae possess hemochorial placentation, and as such, syncytiotrophoblast cells are not formed in their placentas [61,62,63]. Carnivora, including Canidae and Felidae, have two trophoblast layers, with the first layer being syncytiotrophoblast and the second cytotrophoblast. In the ruminants, including Bovidae, syncytiotrophoblast cells are not formed; rather, trinucleate cells, resulting from the fusion between binucleate trophoblast cells and maternal epithelial cell often appear in the uterine stroma [64]. Recently, we [56] and others [55] demonstrated that binucleate and/or multi-nucleate trophoblast cells are derived from the fusion of trophoblast cells. In animals with epitheliochorial placentation, such as horses, camels, pigs, hippopotamuses, and cetaceans, the syncytiotrophoblast layer is not formed; however, binucleate trophoblast cells are found in the horse placenta. Although the placentas serve analogous functions of nutrition and gas exchange, these observations indicate that the cell components of trophoblast cells are far more diversified, which can vary beyond the specific lineages of the mammalian evolutionary tree.

## 5. Requirement of *ERVs* for Placental Evolution

During the last several decades, virologists have occasionally proposed retrovirus’s role in placental evolution. For example, Haig (2012) argued that the placenta became a mammalian tissue in which retroviral genes were ‘domesticated’ to serve an adaptive function in the host [65]. Such an interplay may have contributed to evolutional mechanisms associated with genomic imprinting.

### 5.1. PEG10 and PEG11/RTL1 Are Required for the Acquisition of Placenta

It has been generally accepted that viviparous therian mammals, the marsupials and eutherians, emerged 166 million years ago (MYA) by diversification from oviparous monotremes. Eutherians with chorioallantoic placenta then diverged from marsupials with choriovitelline (yolk sac placenta) 148 MYA [66]. Phylogenetic analysis indicates that occasional endogenization events of LTR of retrotransposons, including ERVs, occurred throughout mammalian evolution. It was found that *PEG10* and *PEG11/RTL1* are paternally expressed imprinted genes [67]. It was then hypothesized that earlier integration of *PEG10* was necessary for the formation of primitive placenta in therians [32,67,68], while the basic structures and functions of the chorioallantoic placenta were established through the endogenization of *PEG11/RTL1* [33,69] and *LDOC1/SIRH7/RTL7* [34]. The formation of primitive chorioallantoic placentas was followed by facilitated structural diversity throughout mammalian evolution in each lineage of eutherians by endogenization of *syncytins* (see next section)*. PEG10* and *PEG11/RTL1* encode proteins that exhibit homology to GAG and POL proteins of sushi-ichi retrotransposon, while *LDOC1/SIRH7/RTL7* encodes only GAG-like proteins of the same retrotransposon. As sushi-ichi retrotransposon is an LTR retrotransposon belonging to the group of gypsy retrotransposons [70,71], it is highly probable that these three genes are derived from some extinct retroviruses [72,73].

The necessity of *PEG10* for placental formation was demonstrated through a gene ablation study in mice, which exhibited embryonic lethality at 10 dpc due to a lack of labyrinth and spongiotrophoblast layers [32]. These observations indicated that *PEG10* plays an essential role in the differentiation and possibly maintenance of trophoblast cells in these layers. *PEG11/RTL1* is essential for maintaining placental fetal capillaries [33,74]. It is expressed in endothelial cells of the fetal capillaries, presumably protected from the attack of surrounding trophoblast cells where *PEG10* is expressed. Therefore, its gene-ablated mice exhibited middle to late embryonic lethality due to growth retardation, resulting from the destruction of the fetal capillary network. Interestingly, PEG10 protease mutants also exhibited perinatal lethality due to severe impairment of the placental fetal capillary network, demonstrating that together with *PEG11/RTL1*, *PEG10* is also crucial for the maintenance of fetal vasculature [75]. Therefore, two ERV-derived genes play essential roles in the establishment of the eutherian type of placenta, as a therian-specific gene (*PEG10*) and a eutherian-specific gene (*PEG11/RTL1*), respectively.

### 5.2. Requirement of LDOC1/SIRH7/RTL7 for the Maintenance of Placental Function

It was demonstrated that another GAG-like protein of the sushi-ichi retrotransposon, LDOC1/SIRH7/RTL7, is essential for endocrine regulation of the placenta [34]. It has long been thought that in rodents, the ovary is the major organ producing progesterone (P4) throughout the course of pregnancy [76]. It was found, however, that the mouse placenta also has P4 transiently during days 9.5–12.5 [34], when the ovarian P4 production exhibits an abrupt fall due to a shift from the corpus luteum of pseudopregnancy to pregnancy [76,77,78]. LDOC1/SIRH7/RTL7 gene-ablated mice exhibited overproduction of placental P4, concurrent with the excess production of placental lactogen 1 (PL1), the latter of which was associated with a delayed change from PL1 to PL2 production in trophoblast giant cells (TGCs) and spongiotrophoblast cells from the junctional zone during the last half of gestation [79]. It was also demonstrated that its gene-ablated mothers exhibit delayed parturition and do not take care of their pups, although these pups are viable when raised by foster mothers [34]. Although further research is required to definitively prove the source of P4 production, these data have clearly demonstrated that not only the ovary but also the placenta are the sources of P4 production in rodents. Therefore, with therian-specific PEG10 and eutherian-specific PEG11/RTL1, LDOC1/SIRH7/RTL7 is also required to maintain mammalian placentas as a eutherian-specific gene.

## 6. Placental Diversity Requires Syncytin ERVs

Mammalian placentas are extraordinarily diverse in terms of morphology, beyond the influence of therian *PEG10* and eutherian *PEG11/RTL1* and *LDOC1/SIRH7/RTL7* genes alone. Despite the statistical improbability of *ERV* integration, the phylogenetic record shows multiple independent instances of these genes’ entry into disparate clades over the last 50 million years, which implicate the *ERV* family genes as prime candidates for the emergence of structural diversification of mammalian placentas. In the human genome alone, remnants of 18 endogenous retroviruses can be found, 16 of which have full coding *env* genes [80]. Only two of these *env* structures possess the fusogenic activity and are highly expressed in the placenta [16,80]: *syncytin-1* (*ERVWE1*) and *syncytin-2* (*ERVFRDE1*), belonging to the HERV-W family and the HERV-FRD family, respectively [16]. These proteins encompass 538 amino acids containing surface (SU) and transmembrane (TM) subunits. They are cleaved by a furin protease, and their interaction is required for their fusogenic activity [81,82]. The location and period of *syncytin-1* and *-2* transcript expression differ: *syncytin-1* mRNA, localized to syncytiotrophoblasts, is highly maintained throughout the gestational period, whereas *syncytin-2* transcripts are limited to cytotrophoblasts and decline in late pregnancy [83]. The syncytin-1 receptor, RD114/mammalian type D retrovirus receptor (ASCT2), is expressed mainly in cytotrophoblasts in the placenta [84], whereas the syncytin-2 receptor, the major facilitator superfamily domain containing 2 (MFSD2), is expressed in syncytiotrophoblasts [85].

Phylogenetic analysis suggests that the *syncytin-2* gene and its fusogenic function were passed on from Platyrrhini to humans (>87.9%), indicating its integration before the divergence of lineage and placing a lower bound on its age at upwards of 40 million years ago [86]. *Syncytin-1*, however, exists only in Catarrhini, not in Platyrrhini, which indicates the more recent date of infection of 25 MYA, after the separation from the Old World monkeys [87]. These observations suggest that *syncytin-2* entered anthropoid lineages and acquired fusogenic as well as immunotolerance activity in cytotrophoblasts, generating the syncytiotrophoblast cell layer. *Syncytin-1* then entered Catarrhini, upon which it may have taken over the fusogenic activity in these lineages. Gestation in Platyrrhini and Old World monkeys, which do not possess *syncytin-1* or its mutated variant, is considerably shorter than in Catarrhines. In humans, syncytiotrophoblasts form from the fusion between cytotrophoblasts approximately 7 to 11 days into gestation and maintain this activity throughout the pregnancy [88]. It should be noted that syncytiotrophoblast cells do not proliferate, ensuring cellular activity can be maintained only through fusion with cytotrophoblasts.

In ruminants, syncytiotrophoblast cells are not found in their placentas; however, bi-, tri-, or multinucleated trophectodermal cells are instead formed, of which cell fusion starts around the time of conceptus implantation and continues until the end of pregnancy. Among the numerous *ERV*s characterized, *syncytin-Rum1* [18] and *BERV-K1* [37] exhibit fusogenic activity. *Syncytin-Rum1*, found in cattle, sheep, and goats, entered ruminants’ genomes 20 MYA, whereas *BERV-K1*, found only in cattle, entered cattle genomes 11 MYA [19]. Although *BERV-K2* and *BERV-P env*s were also reported to be expressed in the bovine placenta, these did not show fusogenic activities [36,37]. In addition, we also reported that *BERV-K3 gag* was expressed in the bovine placenta, of which expression is induced by a WNT agonist [38]. The molecular function of these three *ERV*-derived genes is still unclear, but they could function for placental development in the bovine species.

## 7. Molecular Mechanisms Regulating the ERV Expression and ERVs Serving as Transcriptional Regulators

Both *syncytin-1* and *syncytin-2* genes require a transcription factor, glial cell missing factor homolog 1 (*GCM1*), for their transcriptional regulation. The GCM family genes, regarded as master regulators, are present even in Drosophila [89]. A multispanning transmembrane protein, CD9, is involved in the invasive behavior of cancer cells and cell fusions between sperm and egg as well as myoblasts in muscle cell development [90,91]. CD9 was found to upregulate the *GCM1* and *syncytin-1* genes [92]. *CD9* mRNA and protein expressions increase in BeWo cells when the cells are treated with forskolin. In addition, CD9 was downregulated by a protein kinase A (PKA) inhibitor, suggesting that GCM1 expression is regulated through the cAMP/PKA signaling system [92].

Through the analysis of transgenic mice carrying a 180-kb human bacterial artificial chromosome DNA that contains the full length of corticotropin-releasing hormone (*CRH*) and extended flanking regions, including the endogenous retroviral long terminal repeat transposon-like human element 1B (THE1B), Dunn-Fletcher and colleagues provided strong evidence that anthropoid primate-specific retroviral element THE1B controls placental *CRH* expression [93]. In syncytiotrophoblasts, the regulation of *CRH* by THE1B is through a transcription factor, distal-less homeobox 3 (DLX3).

In addition to *ERV* gene regulation by transcription factors, unique RNA elements control *ERV* RNA processing and translation. A specific sequence motif, termed syncytin post-transcriptional regulatory element (SPRE), was originally found in *syncytin-1* and *syncytin-2* [94]. The SPRE is also present in different *syncytin* family genes, such as *mac-syncytin-3* of macaque, *syncytin-Ten 1* of tenrec, and *syncytin-Car1* of Carnivora. Still, this motif is not found in any exogenous viral sequences [95]. Through a reporter assay experimentation, Kitao et al. concluded that the SPRE may have facilitated the replication of ancient retroviruses and still support the expression of co-opted *ERV* genes in host genomes [95].

Accumulated evidence indicates that *ERV*s are involved in mammalian placentation. It is probable that those integrated into the mammalian genome may work in biological systems other than placentation. For example, one study showed that the transcriptional network of proinflammatory cytokine interferon-gamma (*IFNG*) could have been shaped by ERVs [96]. This involves lineage-specific *ERV*s, which had dispersed their numerous IFN-inducible enhancers in mammalian genomes. For example, a gammaretrovirus, *MER41*, was endogenized in the genome of an anthropoid primate ancestor 45-60 MYA with 7190 LTR elements, from which six subfamilies (*MER41A*, *B*, *C*, *D*, *E*, and *G*) are found in the human genome. *MER41A* is located 220 bp upstream of the gene *Absent in Melanoma 2* (*AIM2*), one of the interferon-stimulated genes that encodes a sensor for foreign cytosolic DNA and activates an inflammatory response. This *ERV* was found to regulate *AIM2* through IFN signaling in anthropoid primates [96]. Furthermore, the enhancer *MER41A/B* insertions regulate dozens of lineage-specific serum response factor-binding loci, including one adjacent to *FBN2*. In humans, an increase in *FBN2* expression leads to the production of the peptide hormone placesin, which stimulates glucose secretion and trophoblast invasion [97]. These observations indicate that ERV-derived enhancers have facilitated the rapid diversification of placentas in mammals.

## 8. Baton Pass Hypothesis: Successive Integration of ERVs

In primate evolutions, including humans, syncytin-2 entered the lineages approximately 40 MYA, followed by syncytin-1 integration 25 MYA. Similarly, in Bovidae, syncytin-Rum1 integrated into the bovine genomes 20 MYA, followed by BERV-K1 (Fematrin-1) 11 MYA. In fact, syncytin-Rum1 is found not only in the bovine subfamily but also in sheep and goats, whereas Fematrin-1 is found only in the bovine subfamily and not in sheep or goats. It appears that during evolution, the exaptation of one *ERV* is not a one-time event but can be just a temporary link in a chain of successive invasions by new *ERV*s. The new interloper retroviral genes may subsume their role, often with beneficial enhancements such as increased efficacy in trophoblast cell fusion, as observed with *Fematrin-1* versus the older *syncytin-Rum1* [18,19]. Meanwhile, the previous *ERV* genes could be lost, but in other cases, they may remain integrated within the genome, repurposed for uses such as immunosuppression as in the case of *syncytin-2* and *syncytin-Rum1* [46,98,99]. These ongoing ERV acquisitions are called “a baton pass hypothesis”, in which a new ERV replaced the preexisting ERV gene and acquired the role that the gene had played [98]. However, these ERVs must be integrated specifically at the relevant locus to be transcribed along with placental genes or through their own LTRs [19,100]. For example, Fematrin-1 is integrated into the intron 18 of FAT tumor suppressor homolog 2 (FAT2), of which gene expression is placenta-specific, and no ERV genes are found around its integration site [19,37]. The baton pass hypothesis is still speculation, but the successive co-option of unrelated ERVs in different species could add to those accumulated, explaining the diversification of placental structures in mammals.

## 9. New Model Explaining Placental Diversity

Fusogenic activity in the mammalian trophectoderm exhibits a great deal of similarity across species, notwithstanding the huge diversity in placental structures and types of placentation such as invasive (humans and murine) or non-invasive (ruminants). Although much research on elucidating placental diversity has been done, molecular mechanisms associated with the construction of structural differences in mammalian placentas have not been definitively characterized. It has been generally accepted that successive integrations of *ERV*-*env* (*syncytin*) genes are the main reasons for placental diversity in mammals (Figure 3). The fact that *syncytin* genes have been independently integrated into the mammalian genomes provides support. However, it should be noted that based on actual experimentation and typical amino acid sequences, their functions are generally limited to fusogenic activity and immunotolerance, which on their own are not sufficient to fully explain the structural diversity of placentas.

Dunn-Fletcher and colleagues demonstrated that the retroviral *THE1B* sequence serves as a cis-element for regulating corticotropin-releasing hormone (*CRH*) gene expression. Further support is expected as genomic and transcriptomic data analytical tools are developed and refined. Recently, progress has been made in research into *ERV* sequences serving as transcriptional and translational regulators [94,95,101,102]. These sequences could be co-opted for newly integrated retroviral gene regulation.

Nevertheless, solid confirmation of a retrovirus integration into sperm or egg has not been obtained, and the mechanism of integration remains unclear. The rarity of such events owes in no small part to the narrow windows of possibility for infection, but conversion to active *ERV*s is also contingent on the perfect confluence of criteria as follows:(a)The insertion of ERVs can make functional genes of the host placenta-specific. i.e., *Fematirn-1* integration into the intron 18 of pregnancy-specific *FAT2* gene;(b)Its own LTR is sufficient to transcribe its own gene segments, which serve as the cis-acting element(s), resulting in the activation of a host gene. i.e., *IFNG*, *THE1B* on *CRH*;(c)It can make use of transcription factors utilized by the pre-existing gene, as per the baton-pass hypothesis. i.e., A transcription factor GCM1 for *syncytin-1* and *syncytin-2.*(d)The ERV is co-opted along with its promoter/enhancer in the integrated genome; i.e., SPRE (syncytin post-transcriptional regulatory element);(e)There is cooperation with miRNAs and/or lncRNAs, yet not definitely characterized under placental/trophectodermal conditions, either alone or together with *ERV*s.

In general, the placentas have lower DNA methylation levels than embryos, allowing free expression of *ERV*s and transposons during gestation, thereby facilitating the selection of advantageous genes from a broader market. Such extraembryonic circumstances might have allowed for not only the domestication of *ERV*s to establish novel endogenous genes via multiple selections but also the dissemination of *ERV*s and transposons throughout genomes as transcriptional regulators. Moreover, *ERV*s could serve as cis- and/or trans-acting factors for functional genes of the host. Similarly, various degrees of maternal-fetal cell interactions in the uterine compartment may [65] have led to changes in kinds and degrees of gene usage, possibly resulting in cellular and morphological changes in placentas. It is interesting to speculate that placentas themselves might have served as an evolutionary laboratory to promote mammalian evolution [72,103].

## 10. Conclusions

Exhaustive research on the elucidation of cellular mechanisms associated with structural diversity of the placenta has continued for several decades. Still, in the last twenty years, new methods such as RNA-seq, iTRAQ, and ChiP-seq analyses have become available to analyze whole transcripts, proteins, and their upstream events to regulate such expression. These advancements have made it possible to isolate molecular events regulating placental diversity and the *ERV*s linked to such variations; however, the mechanisms associated with placental diversity have not been characterized.

It is now clear that the emergence of mammalian placentas was made possible with the acquisition of therian *PEG10* and eutherian *PEG11/RTL1* and *LDOC1/SIRH7/RTL7* genes, followed by independent yet successive integrations of *syncytin*-type genes for structural variations. A question remains whether the placental structures we know now are the ultimate forms or are still evolving. If it is the latter, placental structures may still be diversifying, and new variations await discovery.

## Figures and Tables

**Figure 1 cells-11-02458-f001:**
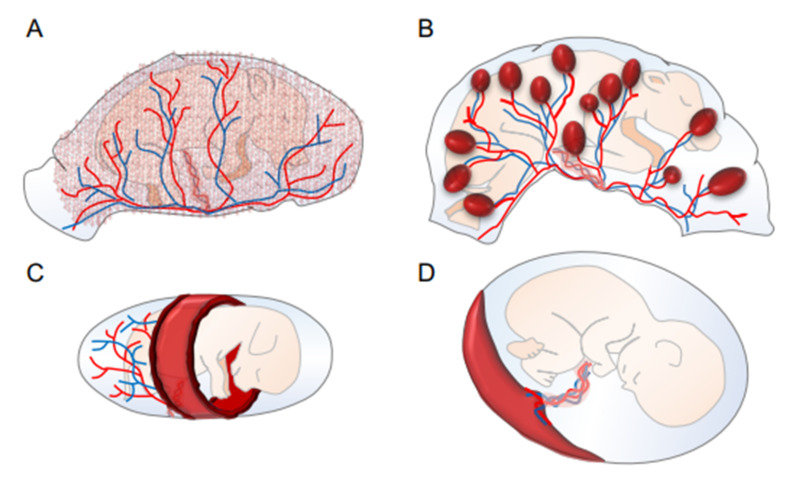
Structural diversity of placentas as classified by the distribution of chorionic villi. Gross anatomy showing diffuse (pigs), cotyledonary (ruminants), zonary (cats and dogs), and discoid (rodents and primates). (**A**): Diffuse placentas have a uniform distribution of chorionic villi covering the chorion’s surface. (**B**): Cotyledonary placentas have numerous, discrete, button-like structures called cotyledons and they join with the maternal caruncle, forming a placentome. (**C**): Zonary placentas have a band-like zone of chorionic villi, of which both edges are primary exchange zones. (**D**): Discoid placentas form a regionalized disc-like structure. Red and blue lines are the artery and vein, respectively, which are joined to an umbilical cord.

**Figure 2 cells-11-02458-f002:**
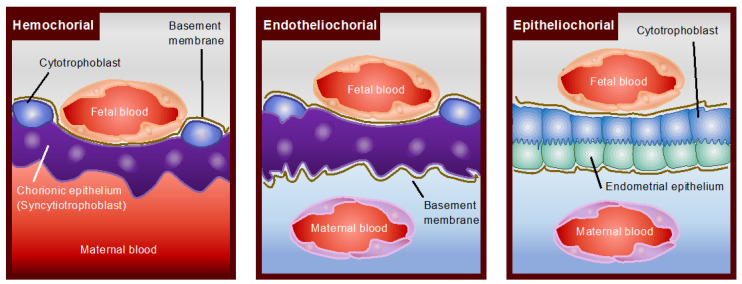
Placental classification according to separation between fetal and maternal blood supplies. There are three layers of separate maternal and fetal blood in Hemochorial (primates and rodents); chorionic epithelium (syncytiotrophoblast), chorionic interstitium (basement membrane), and chorionic capillaries (fetal blood). In Endotheliochorial (dogs and cats), there are five layers between the maternal and fetal blood; endometrial capillaries (maternal blood), endometrial interstitium, chorionic epithelium, chorionic interstitium and chorionic capillaries. In Epitheliochorial (pigs and horses), six layers separate the maternal and fetal blood; endometrial capillaries, endometrial interstitium, endometrial epithelium, chorionic epithelium, chorionic interstitium, and chorionic capillaries.

**Figure 3 cells-11-02458-f003:**
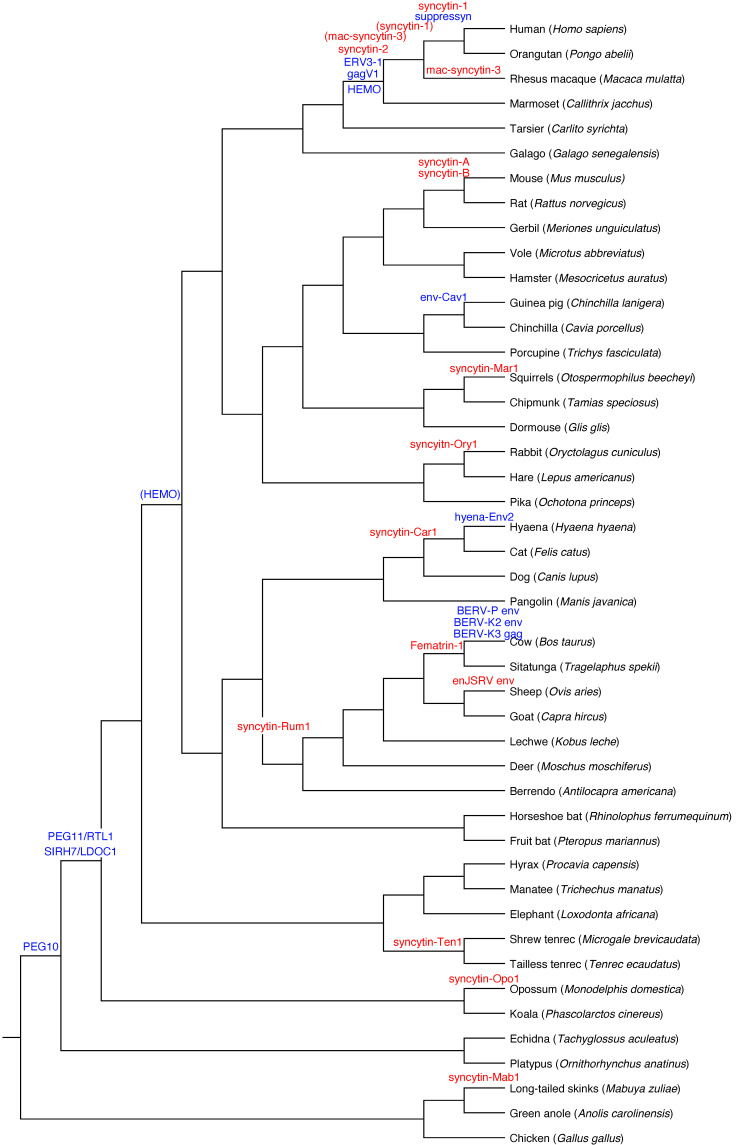
Multiple acquisitions of ERV-derived genes expressed in placenta. The dendrogram indicates a taxonomic relationship of 45 species, including 42 mammals, two reptiles, and one bird. The phylogeny was obtained using the TimeTree website (http://www.timetree.org, accessed on 2 July 2022 [14]). ERV-derived genes expressed in the placenta are illustrated at the location of the presumed inserted lineage. The red or blue color indicated whether the ERV-derived genes were reported to have fusion activity or not, respectively. A gene in parentheses indicated the time of insertion where the gene without parenthesis is when it has acquired function. The details of each ERV-derived gene shown in the figure are summarized in Table 1.

**Table 1 cells-11-02458-t001:** *ERV*-derived genes expressed in the placenta (related to Figure 3).

Gene Name	LTR/ERV	Taxonomy	Fusion Activity	Reference
syncytin-1	HERV-W	Hominidae (Catarrhini)	yes	[15]
syncytin-2	HERV-FRD	Simiiformes	yes	[16]
syncytin-A	-	Muridae	yes	[17]
syncytin-B	-	Muridae	yes	[17]
syncytin-Rum1	-	Ruminantia	yes	[18]
Fematrin-1	BERV-K1	Bovinae	yes	[19]
enJSRV env	endogenous JSRV	Caprinae	yes	[20]
syncytin-Mar1	-	Squirrel	yes	[21]
syncyitn-Ory1	-	Leporidae	yes	[22]
syncytin-Opo1	-	Monodelphis	yes	[23]
syncytin-Car1	-	Carnivora	yes	[24]
syncytin-Mab1	-	Mabuya	yes	[25]
mac-syncytin-3	ERV-V2	Simiiformes	yes	[26]
syncytin-Ten1	-	Tenrecidae	yes	[27]
HEMO	MER34	Simiiformes (Boreoeutheria)	No	[28]
ERV3-1	HERV-R	Simiiformes	No	[29]
suppressyn	HERV-Fb1	Catarrhini	No	[30]
gagV1	HERV-V1	Simiiformes	No	[31]
PEG10	sushi-ichi	Theria	No	[32]
PEG11/RTL1	sushi-ichi	Placentalia	No	[33]
SIRH7/LDOC1	sushi-ichi	Placentalia	No	[34]
env-Cav1	-	Caviomorpha	No	[35]
BERV-P	BERV-P	Bovinae	No	[36]
BERV-K2	BERV-K2	Bovinae	No	[37]
BERV-K3	BERV-K3	Bovinae	No	[38]
hyena-Env2	*-*	Hyaenidae	No	[39]

## Data Availability

Not applicable.

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
