# Peer review of "Endogenous Retroviruses and Placental Evolution, Development, and Diversity"

_cells, 2022, doi:10.3390/cells11152458_

Round 1

Reviewer 1 Report

19thJuly, 2022

Review of Manuscript ID cells-1834961, by K. Imakawa et al., entitled: “Endogenous retroviruses and placental evolution, development, and diversity that is intended for publication in Cells

A knowledge about the evolution and differentiation of mammalian placental structure, despite many years of research in this area, is still incomplete and does not explain the cellular mechanisms involved. The manuscript contains important findings for elucidating the mechanisms involved in the issue of mammalian placental structure differentiation. The Authors highlight the advances in the study of the sequences of endogenous retroviruses (ERVs), which act as transcriptional and translational regulators. These advances seem to be possible due to the use of modern molecular tools, thereby bringing closer the possibility of elucidating the cellular mechanisms involved in the differentiation of mammalian placental structure. The Authors attribute a special role to the discovery of the retroviral syncytin-1 and syncytin-2 genes/transcripts. The latter are the representatives of retrotransposons or retroviral sequences able to move across genomes and induce the fusion of single cells into a syncytium structure. Moreover, these retroviral sequences may be also involved in the processes of cell differentiation within placental tissues. It is important to emphasize the interesting interpretation of previous achievements and the presentation by the Authors of a new model aimed to explain taxonomic or interspecies placental diversity. However, the Authors point out that the processes of integration of retroviral segments with the host genomes appear to be insufficient for their activation; certain criteria/conditions require to be recognized more precisely and comprehensively.

Generally, the paper is very interesting, well written in English and enriched with the Figures  showing, among others, a summary of ERV-derived genes that are expressed in the placenta. This, in turn, is extremely useful for assessing the state of the art. Furthermore, the graphical visualization thoroughly illustrates the discussed contents. The manuscript has been prepared in the format compatible with the requirements of Cells.

In conclusion, I recommend the Editorial Board of Cells to accept the manuscript for publication in its present form. 

Author Response

Thank you very much for your thorough evaluation of our manuscript. Actually, four reviewers evaluated the manuscript and we corrected as much as we can. We sincerely hope that the revised manuscript reads better than the originally submitted one.

Reviewer 2 Report

The paper is a review of endogenous retroviruses and placental evolution and offers an up-to-date review of recent work on the roles of coopted retroviral genes in placental development. It has one glaring gap, but this gap is not unique to the review, but characterizes much other work in the field. This aporia is the failure to acknowledge the role of antagonistic co-evolution in placental evolution. Since Robert Trivers’ paper on parent-offspring conflict in 1975 it has been understood by those who have put the effort into understanding the theory that offspring and parental genes evolve at partial cross-purposes in nutrient exchanges between the generations. This is a fundamental aspect of placental evolution and is a major force driving the diversification of placental structures. (The absence of discussion is a little like ignoring the theory of gravity in a discussion of planetary motion.) Part of the problem is the failure to understand the distinction between proximate and ultimate explanations to use the terminology of Ernst Mayr (1961).

The close formal analogy between theories of parent-offspring conflict and segregation distortion are set out on the first page of

Haig D (1996) Gestational drive and the green-bearded placenta. Proceedings of the National Academy of Sciences, USA 93: 6547–6551.

The authors do not need to cite that paper but they should at least try to understand its first page.

Since the current manuscript purports to be a review of endogenous retroviruses and placental evolution, it should at least engage with the arguments of

Haig D (2012) Retroviruses and the placenta. Current Biology 22: R609–R613.

Author Response

It was not our intention not to include Haig’s paper. In fact, we evaluated PNAS paper, Current Biology and others who contributed to this area of study. We sincerely hope that the inclusion of Haig is sufficient to describe what was going on previously. 

Reviewer 3 Report

This review by Dr. Imakawa and colleagues provides a rather unique view on the evolution and divergence of the placenta correlated with the integration and functional diversification of endogeneous retroviruses (ERVs). As such, it provides a rarely provided perspective of placental development and function.

The article would benefit from an additional figure that depicts the various types of placentation in relation to the interface to maternal blood. For a reader outside the field, the description of ‘synepithelial’ etc will likely remain obscure. This should be integrated with the function of retroviruses in the formation of some of these highly specialized cell types, i.e. syncytial, binucleate, trinucleate etc. Are Syncytin-like elements also required for the formation of bi- and trinucleate cells? Adding this to the figure will substantially facilitate the readability and accessibility of this review to a broader audience.

It does not become clear in this text that LTRs serve as major enhancer and promoter elements, driving alternative transcript isoforms specifically in trophoblast (perhaps due to their hypomethylated status). Therefore, the bullet point list (lines 367-378) should rephrase point 1: the insertion of ERVs does not need to happen close to placenta-specific genes. Rather, the insertion of ERVs can make these genes placenta-specific.

The article will need further editing for English throughout by professional scientific language editing services. Multiple sections are not quite clear or the meaning is skewed.

Line 61: “…chorionic membrane, later a major conceptus side of placenta.” This does not make sense.

Line 96: “When cell cycles of trophectodermal cells are restricted, these cells form giant trophoblast cells through endoreduplication as in the murine species.” This statement needs refinement. Giant cells arise through endoreduplication, which means DNA synthesis takes place in the absence of karyo- and cytokinesis. It is not a general “restriction of the cell cycle”.

EMT does not always precede cell fusion. In fact, in humans syncytiotrophoblast cells form prior to cytotrophoblast cell columns and EVTs, which is where the major EMT process takes place. Rather, EMT is generally coupled to the acquisition of migratory and invasive properties.

Paragraph 124-131, describing data from the authors’ own lab, this section is not clear. Please rephrase.

Lines 143, 144, placental lactogens and PSGs are, at least in the mouse, not predominantly produced by syncytiotrophoblast, but by the junctional zone and giant cells. Please clarify this statement.

Lines 150-151, should this read the “The layer closest to maternal blood”? Cytotrophoblast cells are not facing the maternal decidual stroma. Instead, in humans, it is SynT cells that form that feto-maternal interface until such a time when EVTs arise and invade. It is not clear what is meant here. Since it refers to Muridae, it would seem that this sentence refers to the interhaemal barrier (i.e. sinusoidal giant cells and the two syncytial layers), in which case the sentence is incorrect.

Author Response

Please evaluate the attached document.
